# Human Brain Organoids-on-Chip: Advances, Challenges, and Perspectives for Preclinical Applications

**DOI:** 10.3390/pharmaceutics14112301

**Published:** 2022-10-26

**Authors:** Héloïse Castiglione, Pierre-Antoine Vigneron, Camille Baquerre, Frank Yates, Jessica Rontard, Thibault Honegger

**Affiliations:** 1NETRI, 69007 Lyon, France; 2Sup’Biotech/CEA-IBFJ-SEPIA, Bâtiment 60, 18 Route du Panorama, 94260 Fontenay-aux-Roses, France; 3Sup’Biotech, Ecole D’ingénieurs, 66 Rue Guy Môquet, 94800 Villejuif, France

**Keywords:** brain organoid-on-chip, predictive human based in vitro models, standardization, reproducibility, neurotoxicity

## Abstract

There is an urgent need for predictive in vitro models to improve disease modeling and drug target identification and validation, especially for neurological disorders. Cerebral organoids, as alternative methods to in vivo studies, appear now as powerful tools to decipher complex biological processes thanks to their ability to recapitulate many features of the human brain. Combining these innovative models with microfluidic technologies, referred to as brain organoids-on-chips, allows us to model the microenvironment of several neuronal cell types in 3D. Thus, this platform opens new avenues to create a relevant in vitro approach for preclinical applications in neuroscience. The transfer to the pharmaceutical industry in drug discovery stages and the adoption of this approach by the scientific community requires the proposition of innovative microphysiological systems allowing the generation of reproducible cerebral organoids of high quality in terms of structural and functional maturation, and compatibility with automation processes and high-throughput screening. In this review, we will focus on the promising advantages of cerebral organoids for disease modeling and how their combination with microfluidic systems can enhance the reproducibility and quality of these in vitro models. Then, we will finish by explaining why brain organoids-on-chips could be considered promising platforms for pharmacological applications.

## 1. Introduction

Neurological disorders including Alzheimer’s disease, Parkinson’s disease, Amyotrophic lateral sclerosis, stroke, and brain injuries represent a significant burden for society, and affect up to one billion people worldwide, globally irrespective of sex, age, and education. They are currently a leading cause of disability, and the second largest cause of mortality in the world, with 9 million deaths per year [1]. Yet, no effective treatment exists for many of these disorders. In addition, neurology is one of the most failure-prone areas in the drug development pipeline, despite considerable investment [2]. These high drug attrition rates suggest the limitations of current experimental tools leading to clinical applications. Indeed, in vivo models have failed to translate into any noteworthy advances that could help in the discovery of treatments for neurological disorders. One of the potential reasons for this failure is the gap between rodent models and humans [3].

Modeling the brain remains an elusive challenge because of its inherent complexity, a major factor substantially hampering progress. This complexity starts at the cellular level as follows: the human brain is composed of a cellular heterogeneity with approximately 170 billion cells, organized in an intricated network with 86 billion neurons, and 85 billion non-neuronal cells [4,5] that comprise glial cells—among which microglial cells, resident immune cells of the central nervous system—and endothelial cells [6]. In addition, neurons are organized into a complex neuronal network, estimated to be composed of nearly 10^14^ synapses, with approximately 7000 synapses per neuron in the neocortex [7], ensuring neuronal communication through chemical and electrophysiological signaling. Another parameter that handicaps this incomplete understanding is brain plasticity, the role of which remains elusive but can result in neuronal network reorganization, in strengthened or depressed synapses, or even in changing the functions of a given neuronal population. However, the mechanism behind the occurrence of this plasticity remains open to question [8]. Notably, the brain is considered to be an immune-privileged tissue, which means that the immune responses are tightly controlled and regulated [9]. Such a property is beneficial in the protection of brain cells from immune response-mediated damage but also complicates the predictability of brain responses to drugs. Another level of complexity linked to this immune-privileged tissue arises with the blood-brain barrier (BBB), formed by the vasculature enwrapped in astrocytic endfeet [10]. The unique ability of the BBB to filter the blood selectively poses challenges for drug delivery into the brain, drug candidates’ distribution, and neurotoxicity evaluation. In the context of drug discovery and drug screening, it, therefore, appears necessary to be able to model in vitro the BBB functions [11,12]. Overall, this complexity of the human central nervous system complicates the development of relevant and predictive in vitro preclinical models of human pathophysiology.

Each model has its own advantages and limitations as follows: conventional two-dimensional (2D) cell cultures are simplistic and cost-effective, but the information may be far removed from human physiology as they lack three-dimensionality (3D); 3D cell culture systems are more appropriate to model complex functions of the brain in vitro, but they lack reproducibility; finally, microfluidic devices can recapitulate physiological environments under controlled flows, thus enabling predictive in vitro models [11].

In the last decade, various research groups in academia have developed pluripotent stem cells (PSCs)-based protocols to generate 3D, multicellular, cerebral organoids (for a review of existing protocols, see [13]). Their use in modeling brain biology, early neural development, and human-acquired and genetic diseases has provided new insights into cerebral organogenesis, functions, and neurological disorders, including microcephaly, autism, Parkinson’s disease, and Alzheimer’s disease [14]. However, the adoption of organoid technology for large-scale drug screening in the industry has been hampered by challenges associated with reproducibility, scalability, and translatability to human diseases. Organ-on-chip (OoC) technology has become more widely used in recent years due to its ability to mimic physiological conditions in an in vitro setting [15]. The combination of cerebral organoids and microfluidic systems, named brain organoids-on-chips, could accelerate pharmaceutical testing compared to animal models and 2D cultures by meeting the expectations of both the pharmaceutical and biotechnological industries.

Regarding terminology, cerebral organoids can also be termed neural organoids [16]. Neural organoids are employed as a general term to describe organoids that comprise diverse cell types belonging to several regions of the central nervous system. Cerebral organoids refer to organoids that are mainly composed of cell types from the cerebrum (for a detailed commentary on nomenclature, see [16]). When cerebral organoids are coupled with microfluidic devices, the term brain organoid-on-chip is predominantly found in the literature as a reference to the nomenclature in the field of OoCs, where systems generally take the name of the modeled organ (e.g., brain-on-chip, lung-on-chip, liver-on-chip). In this review, we will examine the advantages offered by cerebral organoids as complex 3D in vitro models and discuss how microfluidic systems can address some of their associated challenges. Subsequently, we will provide an overview of what has already been performed in the recent field of brain organoids-on-chips, discussing their benefits and limitations regarding pharmacological applications. We will then address the current challenges associated with brain organoids-on-chip technologies, with a view to their possible future application with preclinical in vitro models.

## 2. Cerebral Organoids: Advantages, Limitations, and Microfluidic Technology as a Solution

Independently of the cell types used, the two-dimensional nature of conventional cell culture limits the relevance of such models. Indeed, cell cultures grown in 2D cannot recapitulate the in vivo spatial organization of tissues, with 3D architecture enabling cell–cell and cell–matrix interactions, which play a central role in cell morphology, polarity, and gene expression [17]. In addition, cells cultured in 2D conditions are directly in contact with plastic substrates or adherence coating substrates, which biases their interactions with their microenvironment, as they do not reproduce the extracellular matrix (ECM) characteristics [18]. Moreover, in conventional 2D cultures, the friction between liquid and cell membranes, aggravated by medium renewal and handling of culture plates, induces non-physiological mechanical forces such as shear stress, which can affect cell division, morphology, and can ultimately lead to cell death [19].

### 2.1. Cerebral Organoids: Promising In Vitro Models of Human Brain Organogenesis

Cerebral organoids are self-assembled and self-organized cellular aggregates in 3D, obtained in vitro by neural differentiation of PSCs. They comprise different cell types observed in the developing human brain, including neural progenitors, neurons, and glial cells. Contrasting with traditional 2D cell cultures and neurospheroids, cerebral organoids assemble into cellularly complex 3D architectures resembling certain regions of the human brain. These cerebral regions and domains can comprise, in particular, the forebrain, hindbrain, midbrain, cortex, hippocampus, and choroid plexus [20]. Cerebral organoids recapitulate the developing human brain not only at the cellular level but also regarding organizational levels as well as in global developmental trajectories. The cell–cell interaction enables recapitulation of the cellular microenvironment in a more realistic manner by promoting exchanges of information between cells and with the ECM as well as by improving cell differentiation [21].

The 3D cellular models can recapitulate organogenesis aspects of the human brain more accurately than 2D cell culture models. Human brain organogenesis is a complex process involving spatially and temporally regulated events that enable neurodevelopment [22]. Cerebral organoids achieve the ability to recapitulate some of the key aspects of brain organogenesis, from neurogenesis, neuronal differentiation, and migration, to neuromorphogenesis and synaptogenesis. Cerebral organoids generally display discrete proliferative layers reminiscent of the embryonic cortex, including a ventricular zone-like region distinct from an overlying subventricular zone-like region, which comprises outer radial glial cells and intermediate progenitors. In addition, neural progenitors differentiate, and neurons migrate externally, forming distinct and defined cortical plate-like zones. After several months of culture, these cortical regions generally display some degree of spatial organization with the emergence of separated upper and deep cortical layers [20,22,23]. Post-mitotic neurons and synapses can also be detected, suggesting the occurrence of neuromorphogenesis and synaptogenesis within cerebral organoids [23,24]. Interestingly, molecular analyses have suggested the acquisition of cerebral cellular identities and developmental trajectories within cerebral organoids in a timeframe comparable to that of fetal brain samples [22].

### 2.2. Cerebral Organoids: A Diversity of Methodologies

Several methods for generating cerebral organoids exist, resulting in different types of cerebral organoids. These methods can be classified into the following two main differentiation protocols: unguided and regionalized, both with several variants [13,16].

The unguided methods enable the generation of organoids with a high diversity of neural cells reminiscent of several cerebral regions. They rely entirely on spontaneous morphogenesis and intrinsic differentiation capacities within the organoids towards a global cerebral identity, without orienting the differentiation into a specific cerebral region. The first unguided protocol was established by Lancaster and Knoblich in 2013, following the publication of their article the year before about microcephaly modeling using cerebral organoids [20,23]. Briefly, this protocol starts with the aggregation of PSCs into embryoid bodies (EBs), which are then harvested and cultured with specific factors to provide neural induction. Neural differentiation begins with a primitive neuroepithelial development within the EBs, characterized by lateral expansion and budding of neuroepithelial cells. EBs are then embedded into a droplet of an equivalent of ECM called Matrigel, which acts as a scaffold by providing a structural support to facilitate the organoid expansion in 3D and contains additional growth factors. Cerebral organoids further expand within the Matrigel droplet and develop more complex and mature neural structures [20,23]. The regionalized methodology is an alternative method in which exogenous factors are added during the differentiation process to drive the self-organization, self-patterning, and differentiation of stem cells toward specific cerebral regions [25]. In particular, region-specific protocols have been established for the cerebral cortex [24,26,27,28], midbrain [29], cerebellum [30], hippocampus [31], thalamus [28] and hypothalamus [32]. Unguided methodologies enable the modeling of a global cerebral development, which in return has the disadvantage of generating unique organoids with a high heterogeneity. On the contrary, regionalized methods are considered to generate more reproducible organoids but are limited to a particular cerebral region. Overall, the choice between unguided and region-specific approaches depends on the applications and is often seen as a compromise between the diversity and consistency of the model [13].

### 2.3. Cerebral Organoids: Examples of Applications in Neurological Disease Modeling

Cerebral organoids have emerged as a revolutionary in vitro cellular model of the human brain development and functions, in both physiological and pathological conditions [13]. In addition, they are considered a relevant approach to help reduce animal research and to accelerate the process of drug screening [22]. Furthermore, especially when combined with recent genetic engineering approaches, as well as when derived from patient-specific cells, they enable to model and study of the following diverse neurological disorders: (i) diseases occurring during neurodevelopment, (ii) cancers, and (iii) can also give clues about the pathogenesis of neurodegenerative diseases (for a detailed review, see [14]). Briefly, (i) cerebral organoid models of microcephaly [20] and Trisomy 21 [33,34] have been made using patient-derived iPSCs. Prenatal exposures to toxic compounds have been studied on cerebral organoids obtained from healthy patient-derived iPSCs [35], as well as commercially available human embryonic stem cells (hESC) [36]. Oncologic studies (ii) were performed on cerebral organoids made from commercially available hESC in which some cells were nucleofected to introduce oncogene amplifications and/or tumor suppressor mutations [37]. Finally, (iii) Alzheimer’s disease (AD) pathogenesis has been studied using cerebral organoids originating from patient-derived iPSCs carrying familial mutations of AD [38], iPSC transfected with an episomal plasmid vector to introduce mutated Tau protein (P301S) [39], or iPSC chemically induced by Aftin-5 to secrete Aβ42 peptide [40]. Parkinson’s disease has been modeled based on the LRRK2-G2019S mutation seen in sporadic forms, either from carrier-patient-derived iPSC [41], or iPSC genetically engineered using CRISPR/Cas9 technology [42].

In addition, cerebral organoids open up the way to personalized medicine with the utilization of patient-specific cells [13].

### 2.4. Challenges for the Adoption of Cerebral Organoids in Preclinical Applications

Despite the major advances brought by cerebral organoids, there are still some limitations to these models, which hamper their transfer and adoption by the pharmaceutical industry for neurological disease modeling and drug testing.

A major challenge is the high heterogeneity observed between cerebral organoids even when derived from the same PSCs and cultured in the same conditions. These discrepancies include differences in size and morphology between the organoids, as well as structural and functional variations when further analyzed. Such heterogeneity is mainly due to the stochastic nature of the PSCs differentiation and the organoids’ spontaneous self-organization, which inherently lead to differences in cell type proportions and structural organizations. Moreover, some intraorganoids’ heterogeneity is also observed, with differences in cellular densities and structures within the same organoid [13]. The following non-cerebral structures can sometimes be found within the organoids: the presence of germ layers other than neuroectoderm, such as mesoderm [13,43], and the suboptimal presence of cystic cavities [44]. In addition to this inherent heterogeneity, another level of discrepancy arises from the lack of standard criteria for the generation and culture of cerebral organoids, as well as the differences between protocols implemented by the various laboratories concerned. Moreover, the distinct differentiation methods (unguided and regionalized protocols) also exacerbate the diversity of cerebral organoids. Overall, this heterogeneity induces a lack of robustness, reproducibility, and predictability of the model, which raises issues for transfer to an industrial scale, high-throughput screening (HTS), and testing of potential drug candidates in preclinical phases [45].

Another central limitation is the progressive appearance of a necrotic core at the center of cerebral organoids as they grow, due to the lack of vasculature for an adequate oxygen and nutrient supply [13,46]. In practice, there is a diffusion limit for oxygen and nutrient/waste exchanges from the culture medium, which exists at around 400 µm from the surface of the organoid [47]. Therefore, since the progenitor cells at the core cannot be properly supplied with oxygen and nutrients, they progressively undergo necrosis. This necrotic core prevents further growth, limiting the size of cerebral organoids to up to 4–5 mm in diameter and also impedes their maturation, thus preventing the organoids from recapitulating later stages of human brain development [13].

Similarly to the absence of vascular cells, microglial cells are also often absent from this model due to their mesodermal lineage [45]. This lack of brain immune response modeling in cerebral organoids is likely to lead to issues in preclinical studies for drug screening, drug delivery, and neurotoxicity evaluations.

Finally, even if cerebral organoids recapitulate many key features of early human brain development, not all aspects of neurodevelopment are fully recapitulated. This includes the formation of distinct cortical neuronal layers, gyrification, and the establishment of complex neuronal circuitry [13]. In addition, current cerebral organoid models fail to recapitulate most of the late brain development events, such as gliogenesis and myelination, mainly due to the longer time needed for maturation. Moreover, this is worsened by the absence of microglial cells, which play a significant role in brain maturation by inducing the formation of mature dendritic spines and synapses.

When facing these limitations, microfluidic devices are considered promising alternative culture systems likely to improve overall culture conditions and reduce the heterogeneity of the generated organoids in the context of neuroscience research [11,48].

### 2.5. Microfluidic Systems: Promising Technologies to Tackle Cerebral Organoid Limitations

Microfluidic cell cultures rely on engineering sciences and take advantage of technological processes to adapt to biological questions. Thanks to tightly controlled fluid flows, OoCs are considered to improve culture conditions, especially by reducing the shear stress experienced by cells, improving oxygen supply and distribution, enhancing nutrient/waste exchanges, and by facilitating the implementation of chemical gradients. It has been shown that the ability to modulate these flows has an impact on cell morphology, migration [49], and differentiation, particularly for stem cells [50]. In addition, the controlled flows and microenvironment are also considered to improve the reproducibility of cell cultures by reducing heterogeneity between batches. Regarding organoids, controlling flow may allow an enhanced penetration of the nutrients to the center of the organoids. Indeed, Lancaster et al. recently proposed a way to overcome the lack of vascularization in the cerebral organoids by culturing them in microfluidic devices to facilitate nutrients and oxygen uptake within the cerebral organoids [22].

Another advantage of microfluidic systems arises from the flexibility of possible designs [51]. For instance, a simple design with minimalistic human neural circuits composed of a single chamber in which one cell type is cultured [52], or more complex neural networks allowing co-cultures [53]. In more sophisticated platforms, several chambers on the device can comprise different cell types and can be separated or connected thanks to channels or porous membranes, following a logic of compartmentalization [54,55]. In even more complex designs, several devices can be coupled, to enable the connection of distinct organs, forming multi-organ-on-chips [45].

The following other benefits include the integration of multi-parametric analyses: compatibility with imaging techniques (most microfluidic devices are optically clear, allowing fluorescence assay), and electrophysiological measurements (to monitor cells in a non-invasive approach) [11,12,15,56].

Nowadays, microfluidic devices are perceived as alternative novel platforms to current in vitro models based on 2D cell cultures and in vivo models using animals [11,12,57]. Brain organoids-on-chips could provide valid compromises between physiological relevance and reproducibility (Figure 1).

## 3. Brain Organoids-on Chips: State-of-the Art of Promising Models

Brain organoid-on-chip systems have emerged quite recently as a new field of research. They combine cerebral organoid culture with microfluidic devices to improve culture conditions, physiological relevance, reproducibility, and industrial transferability of cerebral organoids. In the literature, there are currently few articles dealing with human brain organoid-on-chips. They can be classified according to their global architecture and the manufacturing process of the microfluidic device as follows (Figure 2, Table 1):(i).Microfluidic devices with 3D culture areas and channels: composed of 3D cell culture areas, and channels for culture medium flows (Figure 2i);(ii).Microfluidic devices with micropillar arrays: comprising micropillars between which cells are cultured in 3D, from iPSCs seeding, to organoid generation, and further growth and expansion (Figure 2ii);(iii).Microfluidic device with an air-liquid interface: air-liquid integrated to the culture platform (Figure 2iii).

### 3.1. Microfluidic Devices with 3D-Culture Areas and Channels

These devices are composed of compartments dedicated to cerebral organoid culture and connected to channels for culture medium renewal and perfusion (Figure 2i).

In 2018, Wang and colleagues developed an innovative brain organoid-on-chip technology to improve cerebral organoid quality compared to conventional cell culture support [58]. The microfluidic device was designed with a perfusion system to enhance oxygen and nutrient supply to the center of the organoids using cerebral organoids generated by unguided differentiation following the Lancaster’s protocol [20]. Specifically, the device was composed of two culture channels into which the organoids were cultured from the EBs stage and maintained for up to 33 days. The culture channels surrounded a central perfusion channel with a continuous flow of culture medium provided by a syringe pump system, and they were wrapped between two additional medium channels with flows of the medium. The architecture of the device enabled medium flows and facilitated nutrient/waste exchanges between the culture channels and the other channels. This system has proved the possibility of enhancing cellular viability and organoid growth with improved cortical development compared to static culture conditions.

In a consecutive article from the same research team, this brain organoid-on-chip technology served as a model of human brain development to study the effects of prenatal nicotine exposure on neurodevelopment [35]. They have demonstrated that brain regionalization, neuronal outgrowth, and cortical development were disrupted in nicotine-treated organoids, with premature neuronal differentiation, thus suggesting that nicotine exposure impairs neurogenesis in early fetal brain development. This study also highlights that brain organoids on-chip technologies could be adequate models and powerful tools in preclinical in vitro studies.

The main advantage offered by this microfluidic device lies in the perfusion system, both enabled by continuous medium renewal with a pump and by the presence of several channels surrounding the culture channels. The design of the device also seems to improve flow exchanges between the channels and enhance perfusion through the Matrigel in the culture channels. This device also seems adapted to real-time imaging. Moreover, this platform enables the study of a variety of other prenatal exposures as well as neurodevelopment in both physiological and pathological contexts. Nonetheless, this microfluidic system does not permit the culture of culture organoids individually, which can be suboptimal in some pathological studies and regarding drug testing applications. It also requires a syringe pump for continuous medium renewal, which is not an easy-to-use format and which represents a supplementary constraint in laboratories and in industry.

Another potential drawback that could be associated with this system is the fact that the cerebral 3D cellular microenvironment is not entirely recapitulated. Indeed, the unidirectional medium flows established in the channels of the device do not model the complexity of fluid flows in the brain.

To face this issue, Cho et al. have developed a specific microfluidic device coupled with an improved ECM for EB embedding [59]. Notably, their work was based on the following two bioengineering strategies: (i) addition of an ECM specific to the brain (brain extracellular matrix), to induce additional brain-specific cues; (ii) dynamic culture conditions in a microfluidic device, to model the bidirectional cerebrospinal fluid flows in the brain. Regarding the microfluidic device, there are two large open culture chambers per device for the cerebral organoid culture, surrounded by three medium-sized chambers containing culture medium. The different chambers are fluidly connected by vertical microchannels that enable medium flows from one chamber to another. Additionally, a periodic and gravity-driven flow from left to right is created by mounting the device onto a bi-directional rocker. Cerebral organoids are generated following Lancaster’s protocol (unguided differentiation) [20]. In this study, they have demonstrated that this microfluidic device improves oxygen diffusion within the cerebral organoids by using oxygen-sensing nanoparticles (PtTFPP-PUAN). The microfluidic conditions also enhanced the organoids’ viability and growth, reduced their heterogeneity, increased neurogenesis and corticogenesis, as well as accelerated maturation (mature neurons, astrocytes, synapses, and electrophysiological signals) after 60 days of culture, compared to classical culture conditions. Such results were even enhanced when the microfluidic culture was coupled with an embedding in the human brain ECM-enriched Matrigel.

The design of this microfluidic device was adapted to the protocol and the introduction of the organoids being facilitated by a system of open chambers subsequently fitted with lids. Such a design demonstrates the flexibility of microfluidic devices and their adaptability to experimental constraints. Further advantages include the combination of microchannels that connect the medium and culture chambers with a left–right flow of the medium, which seems to play a central role in the improvements observed regarding organoid viability and maturation. In addition, this improved medium flow, renewal, and diffusion through the organoids are obtained with a pump-free system adapted to industrial transfer. Moreover, this platform appears to be adapted to a whole range of studies of brain development and functions, in both physiological and pathological conditions.

Despite the foregoing, this microfluidic device does not seem suitable for the culture of cerebral organoids from their earliest stages. Although EBs could easily be introduced into the chambers, the Matrigel embedding step required in the Lancaster’s protocol does not seem feasible with this format, due to the presence of lateral microchannels that would be blocked by polymerized Matrigel. Organoids should instead be removed from the device, embedded in Matrigel, and repositioned. As for that in previously described articles [35,58], this device does not permit the individual culture of organoids, and this is likely to limit drug testing applications.

The culture of cerebral organoids, their expansion, and maturation are long processes. In the aforementioned article, this point was not addressed since they still required long culture times, up to 120 days, for a distinct separation to be observable between the upper and deep cortical layers (immunostained by Satb2 and Ctip2 respectively).

One potential experimental strategy that could overcome this issue is the culture of cerebral organoids in a device with a constrained environment to accelerate maturation. Karzbrun and colleagues have developed a device with a constrained 3D-culture area to model cerebral wrinkling and folding in cerebral organoids [60]. To this end, they designed a device inside of which the organoid was compressed in a small culture chamber of only 150 µm in height (compared to 8 mm in Cho et al.). The chamber was covered with a semi-permeable membrane and a medium reservoir to facilitate the diffusion of nutrients. The device was composed of different parts, which were progressively assembled to facilitate the successive steps of cerebral organoid culture (for instance, EBs were positioned in an open chamber, which was added to the rest of the device afterward). In this design, a bottom coverslip positioned under the device facilitates in situ imaging during the culture. Interestingly, this platform is also adapted to in situ immunostaining of the organoids. The cerebral organoids were generated following an unguided differentiation but according to a different protocol and timescale than the Lancaster’s method, and with smaller EBs at the initial stages. This study demonstrated the appearance of surface wrinkles in cerebral organoids when grown in a constrained space and enabled the study of a model of lissencephaly. Thus, this device provided, for the first time, a platform to study mechanisms underlying brain folding as well as associated pathologies. In addition, the constrained environment and the resulting surface wrinkles may also imply an accelerated maturation of the organoids compared to non-spatially restrained cultures.

However, the step-by-step assembly of this device does not seem compatible with scaled-up fabrication processes and industrial transfer. Nevertheless, approaches to accelerate the maturation of cerebral organoids are of great interest, especially for some neurological disease modeling and pharmacological studies, and solutions adapted to an industrial transfer should be further investigated.

The inherent absence of vascular cells in cerebral organoids raises another challenge regarding their viability. To answer this issue, Salmon and co-workers developed vascularized cerebral organoids [61] by co-culturing a cerebral organoid with a vasculature composed of human PSCs-derived endothelial cells and pericytes inside a microfluidic device. As for the cerebral organoids, the 3D vascular network was obtained by the differentiation of human PSCs in a 3D culture. The organoids were generated following the Lancaster protocol (unguided differentiation) [20]. The aim of this study was to recapitulate the temporal synchronization and spatial orientation of both cultures in an in vivo-like manner. The microfluidic device is structured with an individual culture chamber for the cerebral organoid culture, surrounded by lateral channels for endothelial cells and pericytes to vascularize the central organoid. In this study, the authors have demonstrated the feasibility of co-culturing cerebral organoids with a 3D vasculature. They also enhanced both perfusion and permeability of the generated vascular network and achieved accelerated maturation of the organoids after 15 and 30 days of culture, compared to control non-vascularized cerebral organoids. This device could also serve as a base on which to model the BBB, with the addition of human astrocytes and pericytes. Finally, considering the fabrication process and given the standardizable format of the device and its suitability for drug testing applications, it would appear to be a likely candidate for transfer to an industrial scale. This device seems also to be adapted for the vascularization of other types of organoids. However, the most optimal flow regime remains to be determined, along with its implications for organoid growth.

All the microfluidic systems described in this part have been fabricated in-house by the experimenters. Globally, a major advantage of this “home-made” strategy is that the devices are specifically designed to answer the needs of the experiments. However, the attendant drawback is that they are not all suitable for large-scale production due to their in-house designs and fabrication methods.

### 3.2. Microfluidic Devices with Micropillar Arrays

Such devices comprise multiple micropillar structures between which cells are seeded, self-aggregated to form EBs, and further expand into organoids (Figure 2ii). Therefore, the main advantage of these micropillar devices is the entirely in situ generation of the cerebral organoids, which reduces the manual transfer of the EBs for the harvesting step.

In the first article from Zhu and colleagues, a device is presented composed of a micropillar array with octagon-shaped micropillars [62]. The protocol followed to generate the cerebral organoids was Lancaster’s methodology [20], with all the steps being performed in situ. Its objective was to design and optimize the micropillar arrays to generate cerebral organoids, especially insofar as the dimensions of the pillars and the distances between them are concerned. The organoids obtained were then characterized to assess correct neurogenesis and to identify different brain structures. Later, the same research team applied this micropillar device to investigate neural impairments induced by cadmium—known to be a neurotoxic compound with a long biological half-life—during early brain development [63]. For the treated organoids, they observed an increased cell apoptosis, impaired neural differentiation, and maturation, compared to the control organoids, suggesting that cadmium exposure impaired neurodevelopment. Therefore, this microfluidic system would seem to be a relevant approach for drug testing and neurotoxicity studies.

Another team developed its own brain organoid-on-chip platform with micropillar arrays and coupled with cortical organoids (regionalized differentiation) to study the effects on the neurodevelopment of prenatal exposure to valproic acid (VPA) [64], and exosomes derived from breast cancer [65]. For the study with VPA, the treated cortical organoids exhibited impaired neurodevelopment with increased neuronal progenitors but inhibited neuronal differentiation and altered forebrain regionalization. Interestingly, similarities with autism patient-derived organoids were observed, highlighting the risk of autism onset associated with prenatal VPA exposure. Regarding the study with breast cancer-derived exosomes, treated organoids displayed not only impaired neurogenesis but also carcinogenesis with activated signaling pathways associated with breast cancer and medulloblastoma.

Overall, these devices with micropillar arrays could be developed into interesting platforms for HTS and drug testing, especially for applications in neurotoxicity studies. However, the absence of compartmentalization between the organoids could be problematic for some studies in which individual responses of the organoids might be required (e.g., measurement of metabolites). Moreover, as yet, the medium flow does not seem fully controlled within the micropillar arrays, indicating a need for further microfluidic studies.

### 3.3. Microfluidic Device with an Air-Liquid Interface

Ao et al. have developed an in-house device made of individual chambers for cerebral organoid culture, with a generation protocol entirely in situ [36]. The chambers contain an air-liquid interface, intended to promote oxygenation within the medium and to minimize hypoxic core formation within the organoids. The microfluidic device also induces a physical restriction, limiting the size of the growing organoids to under 2 mm, thus enhancing their reproducibility in terms of dimensions. The authors demonstrated that the generated organoids recapitulated structural and electrophysiological characteristics of the early human brain while exhibiting reduced hypoxia and being more homogenous in terms of size compared to conventional cultures.

This platform was also used to study neurotoxic effects on cerebral development in prenatal cannabis exposure. Notably, the treated organoids displayed reduced neuronal maturation, impaired neurite outgrowth, and reduced spontaneous firing rate.

Since this device is compatible with commercially available standard 6-well or 24-well plate formats, it seems well adapted to a transfer at a larger scale. Moreover, it enables to simultaneously culture a high number of cerebral organoids (up to 169 in a 6-well plate format) while preventing fusion and merging between them, as opposed to conventional culture conditions. Overall, the main advantages of this system are the reduced hypoxic core formation and the enhanced homogeneity of sizes between the generated organoids. Ultimately, this approach seems to provide an experimental strategy enabling control of the size of organoids. However, size homogeneity between the organoids relies on the measurement of the diameters of the organoids, which provides interesting information but is generally considered less representative than the measurements of volume, since organoids expand in 3D.

Interestingly, Giandomenico and Lancaster have recently described an innovative cerebral organoid culture based on the combination of organotypic slices cultured with an air-liquid interface to improve oxygen and nutrient supply and so accelerate organoid maturation [66]. This system increased neuronal survival and maturation, axon outgrowth, and circuit formation, leading to an active neuronal network. However, in this protocol based on organoid slices, the organoids are not maintained intact during culture, in contrast to the protocol of Ao et al., where whole organoids are cultured at the air-liquid interface thanks to microfluidic technology.

## 4. Discussion: Current Insight towards Industrial Applications of Brain Organoids-on-Chips in the Field of Pharmacology

Proposing human brain-organoids-on-chips as predictive platforms to study drug responses for neurological disorders requires taking into account several factors. These include (i) conceiving methods of obtaining cerebral organoids that closely mimic the physiology of the human brain, with high reproducibility and an increased maturation rate in order to limit the time spent in culture, (ii) having access to standardized and reproducible manufacturing processes, while (iii) increasing performance for high throughput analysis via non-invasive monitoring techniques such as electrophysiology, and (iv) having the capability to render the model more complex by integrating several cell types or forming multi-organ/multi-organoids-on-chip (for an overview of current brain organoids-on-chips advantages and limitations towards pharmacological applications, see Table 2). Since cerebral organoid generation protocols require expertise in stem cells and neurodevelopment, the platform must remain affordable and easily transferable to user laboratories while being standardized and adapted to HTS and automation processes for industrial transfer.

### 4.1. Standardization Methods for Reproducible Cerebral Organoid Generation

The stochastic nature of PSCs self-organization and differentiation inherently leads to heterogeneity and variability between cell types and structures obtained within and between individual cerebral organoids. In particular, unguided differentiation results in higher heterogeneity compared to regionalized methods [13]. Yet only one research team to date has used a guided protocol to generate cortical organoids (compared to eight using an unguided protocol) on a micropillar arrays-based platform [64,65] (Table 1). However, using regionalized protocols seems more suitable for larger-scale pharmaceutical applications due to their inherent reduced heterogeneity.

For pharmacological studies, the following other key variable parameters of the organoid generation protocols should be considered: (i) the source of PSCs, ranging from commercially available PSC lines to patient-derived cells, as well as the quality and the initial state of the PSCs used to generate the organoids, (ii) experimental parameters of cell culture, and (iii) the presence of undefined components.

Culturing PSCs in feeder-dependent conditions is considered to increase the variability and inconsistency of the cells, in addition to being technically more difficult [13]. Therefore, feeder-free cultures appear more appropriate for preclinical studies in terms of reproducibility.

Regarding other experimental parameters, including the number of passages of the PSCs used, their state of confluency before their self-aggregation into EBs, or the initial morphology of the EBs, it seems important to define quality controls and selection criteria for improved standardization and reproducibility. Another important aspect to consider is the limitation of undefined ingredients in the protocols. Notably, concerning Matrigel, commonly used either to embed the EBs [20] or diluted in the culture medium [27]. The animal origin, undefined composition, batch-to-batch variability, and relatively high cost of Matrigel, as well as its recent global shortage, provide reasons to reflect on the necessity of replacing animal-derived ECMs in preclinical studies and on the need to multiply industrial suppliers. Alternatives to Matrigel could include defined non-animal hydrogels. However, the latter must be carefully selected for their mechanical properties since they are known to influence cerebral organoid development [51]. Another already existing solution may lie in the use of cerebral organoid protocols that do not require an ECM [24,26]. Interestingly, in the study of Cho et al., a human brain ECM-enriched Matrigel was used to embed the EBs [59]. They demonstrated that the presence of ECM extracted from human cortex samples led to improvements in the viability and growth of the organoids, as well as an accelerated maturation at structural and functional levels. Human brain ECM would certainly be more physiologically relevant; however, the lack of supply and costs would probably hamper its use in preclinical research. Nonetheless, synthetic hydrogels that model the human brain’s ECM physical structure and proteome could be a promising alternative adapted to the cerebral organoid culture.

Overall, harmonization and standardization of cerebral organoid protocols should be envisaged to facilitate their adoption by the pharmaceutical industry and regulatory bodies. A further consideration is that commercial cerebral organoids may also constitute a potential alternative for pharmacological studies by facilitating standardized generation and culture protocols [14].

### 4.2. Standardization of Microfluidic Fabrication Process for Reproducible Devices

As highlighted in Table 1, microfluidic systems tend to enhance the reproducibility and quality of 3D cell culture. However, microfluidic device implementation requires quite complex and time-consuming prototyping and fabrication processes, and specific equipment. Since the generation of cerebral organoids is challenging and can lead to great variability between batches, the manufacturing processes must therefore ensure robust and reproducible microfluidic architectures. In addition, microfluidic systems need to be easy to use, and suppliers should provide all the details and protocols to be adopted by the whole scientific community [15]. Quality criteria and device standardization are thus necessary to allow comparison and experimental reproducibility between laboratories, and to facilitate large-scale transfer for pharmacological studies. Overall, this necessary standardization could be achieved using commercial microfluidic systems [51]. Similarly, the materials used to fabricate the devices are another important parameter to consider [67].In particular, the absorption properties of the material must be considered, especially in the case of certain small molecules [68].

### 4.3. Compatibility of Microfluidic Technologies with High-Throughput Pharmacological Assays: Need for Relevant Read-Outs and Data Collection Methods

The most common read-outs and associated data currently obtained from brain organoid-on-chip platforms comprise immunofluorescence (IF) staining and reverse transcription-quantitative polymerase chain reaction (RT-qPCR) (Table 3).

Most of the publications reviewed in Table 3 have performed immunostaining characterization on cerebral organoid cryosections, which requires organoid recovery from the microfluidic device (Table 3, [59,64,65]). Organoid recovery and subsequent slicing do not represent a suitable solution for high-throughput preclinical studies. Only two studies have immunolabelled the organoids directly within the device (Table 3, [60,61]). Subsequent microscopic observations were permitted thanks to the optical transparency of the microfluidic devices, indicating that particular attention should be paid to the refractive index of the material when manufacturing the device. Karzbrun et al. also implemented the use of fluorescent reporters with their cerebral organoids. For this purpose, they electroporated PSCs with fluorescent reporters for chromosomes or actin, enabling subsequent fluorescent visualization of the organoids’ global cellular structures directly within the device [60]. Contrary to classic IF staining on cryosections, this non-invasive visualization strategy could be adapted to industrial and high-throughput handling, while still being consistent in terms of organoids follow-ups and characterizations in preclinical applications. However, the use of genetically modified cells can induce irrelevances regarding human physiology. In addition, techniques of high-content imaging of cerebral organoids could also be more suitable for industrial-scale and preclinical studies [69], especially by applying automated microscopy coupled with data processing algorithms.

RT-qPCR and RNA sequencing (RNAseq) represent common read-outs among the reviewed articles for transcriptomic analysis of brain-organoids-on-chips (Table 3). In particular, single-cell RNAseq permits the measurement of the transcriptome with a single-cell resolution and has been applied to cerebral organoids [27,70]. However, the required dissociation of the organoids induces the loss of spatial information. Other approaches to spatial transcriptomics overcome this limit by mapping transcriptomic data according to the original localizations within the tissues (e.g., RNA in situ hybridization techniques) [71]. Spatial transcriptomic methods have also been applied to cerebral organoids [27]. Nonetheless, similarly to IF staining, spatial transcriptomic methodologies require organoid sectioning and do not seem entirely adapted to high-throughput preclinical studies. Bulk and single-cell transcriptomic data collection methods such as RT-qPCR and RNAseq require RNA extraction from the organoids. Even if they are also destructive methods, they appear more suitable for brain organoid-on-chip platforms and for preclinical applications, especially when RNA extraction is realized within the microfluidic device, as performed in the article from Karzbrun and colleagues [60].

Overall, read-outs and data collection methods employed with brain organoid-on-chip systems require the removal of the organoid (Table 3). This organoid retrieval does not seem completely optimal for industrially scaled in vitro studies but can be improved by coupling the microfluidic devices with convenient retrieval systems and by developing platforms compatible with batch organoid cultures that can be stopped at different culture timepoints to perform invasive read-outs such as IF staining and transcriptomic analyses. In addition, non-invasive read-outs that leave the organoids unimpaired throughout the culture and are compatible with automation and high-throughput handling should also be considered for pharmaceutical applications. In particular, analyses based on the regular sampling of the culture medium (also called cell culture supernatant) can provide significant insights into viability monitoring and metabolomic studies, particularly in the context of neurotoxicity and pathological studies. Indeed, cytotoxicity, oxidative stress, or pathological biomarkers are released in the culture medium and can be subsequently detected, often by colorimetric, fluorometric measurements or more quantitative methods such as liquid chromatography coupled to mass spectrometry (LC-MS). In the article from Cho et al., glucose and lactate levels present in or secreted from cerebral organoids are directly measured in the culture medium in the microfluidic device [59]. However, due to the small volumes of culture medium present within microfluidic systems, attention should be paid to reproducibility. Another potential non-invasive read-out that can be easily coupled to microfluidic platforms lies in electrophysiological recording using micro-electrode array (MEA) technologies.

Similarly, representative and standardized biomarkers of cerebral organoids should be identified for each read-out and data collection methods performed, to facilitate comparisons between studies. In particular, markers of viability, cytotoxicity, or oxidative stress could be selected in neurotoxicity evaluations, while disease-specific markers should be investigated in disease modeling and drug testing studies.

At an industrial scale, standardized algorithms and machine learning technologies could be suitable for large-scale automatic data processing and analyses [72,73,74].

### 4.4. Compatibility of Microfluidic Technologies with High-Throughput Pharmacological Assays: Electrophysiology as Relevant Non-Invasive Read-Out

A hallmark of cerebral organoids’ functional analyses lies in electrophysiological recording [75,76]. Indeed, mature cerebral organoids have the ability to recapitulate complex neuronal networks with electrophysiological communications that can be monitored. Conventional technologies developed to measure electrical signals of neuronal activities relied on electrodes. However, their invasive implantation is not suitable for long-term and stable recording within cerebral organoids. Currently, the most commonly used methods with cerebral organoids include the following: (i) patch-clamp, (ii) calcium imaging, and (ii) MEA technologies [75]:(i).Patch-clamp: provides high temporal resolution, which enables to monitor responses to pharmacological compounds, but little spatial resolution, which prevents information about neuronal networks’ connectivity within organoids ([24]);(ii).Calcium imaging: enables larger-scale activity information with higher spatial resolution, permitting neural networks analysis within organoids, but with a low temporal resolution, and dependent on imaging capabilities ([20]);(iii).MEA technologies: provide both high temporal resolution and spatial resolution ([66]). In 2D MEA, cerebral organoids are positioned on the MEA support, and action potentials can be detected and recorded from neurons located on the surface of the organoid in contact with the electrodes.

However, these technologies are not compatible with the three-dimensionality of cerebral organoids since they are limited to the organoid surface and prevent recording in deeper layers. An alternative lies in the use of organoid organotypic slices, where slices are electrophysiologically recorded and can be maintained in culture [66,77]. Nonetheless, these techniques imply a loss of organoid structural integrity.

To overcome these limitations, emergent MEA technologies are currently under development, including 3D MEA, to improve electrical detection in the center of cerebral organoids. Such whole cerebral organoid electrophysiological detection seems necessary to obtain information about global connectivity and circuitry within the organoids, as well as in the case of pathological studies involving diseases with synaptic dysfunctions. In addition, MEA technologies are also considered to be more compatible with high-throughput recording by comparison with other methods, especially when coupled with algorithms for data analysis [78]. Another advantage of MEAs regarding pharmacological studies relies on the possibility to assess the electrophysiological activity before and after drug exposure, thus assaying the effects of the studied compound.

Interestingly, technologies based on calcium imaging and using MEA systems are directly compatible with microfluidic devices. Thanks to the optical transparency of the materials used in microfluidic platforms, in situ calcium imaging is feasible. Regarding MEA, microelectrodes can be placed directly at the bottom of the cell culture chambers in the devices, enabling direct recording [56]. For brain organoid-on-chip systems, the combination of microfluidic devices with 3D MEA technologies seems an interesting perspective for a high-throughput non-invasive long-term electrophysiological recording of cerebral organoids, providing a relevant non-invasive read-out adapted to pharmacological studies.

### 4.5. Enhancing Capabilities by Coupling 3D Cell Culture with the Blood-Brain Barrier

One of the major obstacles in the development of efficient therapeutic drugs for many neurological diseases is the high selectivity of the BBB. Developing high-fidelity in vitro models of the BBB is necessary in order to test drug permeability and distribution. Conventional in vitro models are based on transwell culture systems with co-cultures of several neurovascular cells—generally endothelial cells and astrocytes. However, these minimalistic models are limited in terms of physiological relevance. Addressing this issue, microfluidic devices have been proven to recapitulate in the physiological microenvironment the BBB complexity thanks to 3D geometry, the ability to compartmentalize, and improved fluid flows [12]. Typically, a microfluidic-based BBB device consists of a porous membrane separating two channels, thus forming two distinct compartments modeling vascular and neural sides, separated by an endothelial cell monolayer. These in vitro models allow investigation of the ability of different compounds to interact with endothelial cells, pericytes, and astrocytes, and to transit across the BBB [79]. An interesting future perspective could be the combination of cerebral organoids with such BBB models, using complex microfluidic platforms, notably for in vitro neurotoxic assessment.

Similarly, another perspective of brain organoids-on-chips complexification lies in the emerging field of multi-organs/organoids-on-chips, with diverse co-cultures of several cell types or organoids, and the addition of porous membranes to model physiological barriers.

## 5. Conclusions

The synergy between cerebral organoid culture and microphysiological chips, termed “brain organoid-on-chips”, can recapitulate the complexity of the human brain while leveraging the advantages of technology by recreating a physiological and controlled microenvironment. The 3D microfluidic in vitro models based on cerebral organoids will widen the field of study of toxicology (ADME-tox), as well as the delivery and screening of molecules that target the brain under more physiologically relevant conditions. However, routine applications in the early phases of drug development will require reduced variability and increased quality in terms of culture duration and expression of maturation markers to achieve the possibility of high throughput analysis. Thus, the adoption of brain organoids-on-chips for routine applications will require the following:-The establishment of standardized cell culture conditions with defined and reproducible validation and characterization criteria;-Proof of concept that the technology contributes efficiently and reliably to clinical success for novel therapeutics and improves translational research by providing evidence of successfully predicted human responses;-Evidence that the model will reduce the need for animal testing while remaining consistent with the scientific aims of the study.

Once validated, evolution along two axes, each with substantial benefits, might be proposed as follows: firstly, its application to personalized medicine with the use of primary human cells obtained from patients, and secondly, the generation of multi-organoid-on-chip platforms.

## Figures and Tables

**Figure 1 pharmaceutics-14-02301-f001:**
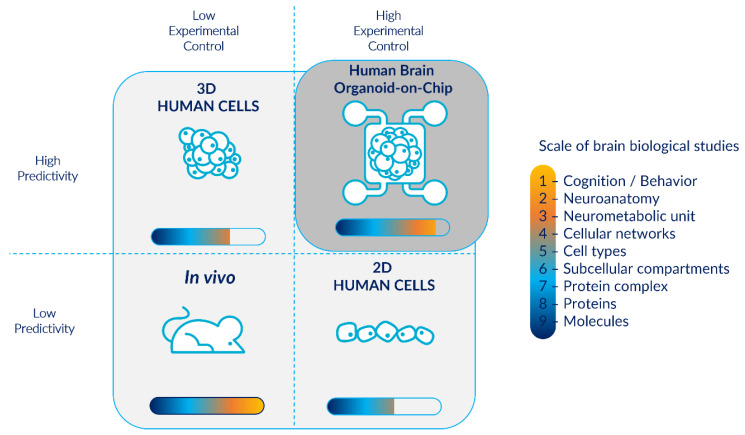
Human brain organoid-on-chips as physiologically relevant and reproducible models for the study of brain biological organization. Even if in vivo models enable a wider diversity of biological neurological studies, they are associated with low predictivity and experimental control. They are also considered as a black box model: often privileged by researchers despite the availability of other models. In contrast, in vitro 2D human cells allow high experimental control, however they have a low predictivity and do not permit brain biological studies at a higher scale than the cellular level. On the other hand, 3D structures such as cerebral organoids enable more complex studies with high predictivity but are associated with lower experimental control and high inter- and intra-batch heterogeneity. The combination of cerebral organoids with microfluidic technology provides a model with high experimental controllability, particularly for the modeling of physiological microenvironments in a minimalist approach, while ensuring a high predictivity, and enabling a wide range of biological studies.

**Figure 2 pharmaceutics-14-02301-f002:**
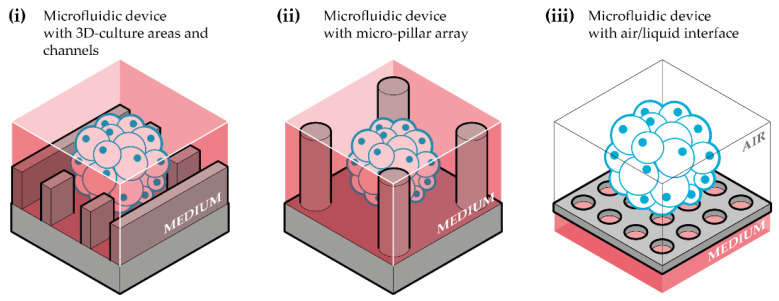
Schematic representation of the three main types of microfluidic devices used for brain organoid-on-chip systems. (**i**) Microfluidic device with 3D-culture areas and channels: device composed of a compartment for cerebral organoids culture, linked to channels for culture medium flow and perfusion; (**ii**) microfluidic device with micropillar array: device that contain micropillars between which cerebral organoids are generated and cultured; (**iii**) microfluidic device with air/liquid interface: device adapted to a cerebral organoid culture in an air-liquid interface.

**Table 1 pharmaceutics-14-02301-t001:** State-of-the-art of recent brain organoids-on-chip technologies with their advantages for cerebral organoid cultures (W: width; L: length; D: diameter; H: height).

Type of Device	Fields of Study	Protocol for Organoid Generation	Characteristics of the Microfluidic Device	Advantages of Microfluidic Technology for Cerebral Organoids Culture	References
Microfluidic devices with 3D culture areas and channels	Neurodevelopmental toxicity: tests of prenatal nicotine exposure effects on neurodevelopment	Unguided (Lancaster’s protocol)	- Culture channels (w: 2.5 mm × L: 14 mm) - Perfusion channel in between - Medium flow channels - Continuous flow (syringe pump)	- Enhancement of cellular viability and growth	[35,58]
	Modeling of the cerebrospinal fluid flow	Unguided (Lancaster’s protocol)	- Culture chambers (D: 8 mm) between medium chambers - Periodic flow: device placed on a bi-directional rocker - Microfluidic device with bi-directional fluid flows	- Enhancement of viability and growth - Improvement of oxygen diffusion within the organoids - Acceleration of the maturation (structural + functional) - Enhanced reproducibility	[59]
	Modeling of cerebral folding and diseased modeling of lissencephaly	Unguided (not Lancaster’s protocol)	- Constrained culture chamber (150 µm height) - Medium perfusion through a semi-permeable membrane between the chamber and a medium reservoir above	- Appearance of surface wrinkles and folding in organoids - Lissencephalic organoids displayed reduced convolutions - Entirely in situ organoids culture	[60]
	Vascularized cerebral organoid in a microfluidic device	Unguided (adapted from Lancaster’s protocol)	- Individual culture chamber (D: 2 mm) per device, surrounded by channels for endothelial cells/pericytes to vascularize the organoid	- Perfusion and permeability of the vascular network - Improved neuronal maturation - Could be used as a model of BBB	[61]
Microfluidic devices with micropillar arrays	Neurodevelopmental toxicity: tests of prenatal cadmium exposure effects on neurodevelopment	Unguided (Lancaster’s protocol)	- Micropillar arrays with octagon-shaped pillars (D: 1 mm × H: 0.8 or 0.6)	- Characterizations of the organoids (neurogenesis and brain structures) - Entirely in situ organoids culture	[62,63]
	Neurodevelopmental toxicity: tests of prenatal exposure effects on neurodevelopment with valproic acid and breast cancer-derived exosomes	Guided (cortical organoid)	- Micropillar arrays (D: 1 mm × 1 mm)	- Entirely in situ organoids culture	[64,65]
Microfluidic devices with air-liquid interface	Neurodevelopmental toxicity: tests of prenatal cannabis exposure effects on neurodevelopment	Unguided (STEMdiff cerebral organoid kit), adapted from Lancaster’s protocol)	- Culture chambers with an integrated air-liquid interface (D: 2 mm)	- Improved viability + reduced hypoxia + enhanced homogeneity of diameters - Entirely in situ organoids culture—Possibility of in situ Matrigel embedding (Lancaster’s protocol)	[36]

**Table 2 pharmaceutics-14-02301-t002:** Advantages and current limitations of brain organoids-on-chip technologies regarding pharmaceutical applications.

Type of Device	References	Scalability	Reproducibility	Maturity *	Functionality **	Drug Permeability (BBB Modeling)
Microfluidic devices with 3D culture areas and channels	[35,58]: Perfusable device with culture channels + perfusion channels Prenatal nicotine exposure	Possible	Yes	33 days	No	No
	[59]: Device modeling cerebrospinal fluid flow + with brain ECM	Possible	Yes	120 days	+/− (Ca^2+^, patch-clamp)	No
	[60]: Device with constrained culture chamber Cerebral folding and wrinkles Lissencephaly modelling	Does not seem possible	Yes	20 days	No	No
	[61]: Device to develop vascularized cerebral organoids	Possible	Yes	30 days	No	+/−
Microfluidic devices with micropillar arrays	[62,63]: Prenatal cadmium exposure	Possible	Yes	40 days	No	No
	[64,65]: Prenatal exposure with valproic acid and breast cancer-derived exosomes	Possible	Yes	70 days	No	No
Microfluidic devices with air-liquid interface	[36]: Prenatal cannabis exposure	Yes	Yes	90 days	+/− (2D MEA)	+/−

* Maximal timepoint. ** Electrophysiological recording assays.

**Table 3 pharmaceutics-14-02301-t003:** Readouts and data collection methods of current brain organoids-on-chip technologies. Abbreviations: IF: immunofluorescence staining, RT-qPCR: reverse transcription quantitative polymerase chain reaction, TUNEL: terminal deoxynucleotidyl transferase dUTP nick end labeling, LC-MS: liquid chromatography coupled to tandem mass spectrometry. RNAseq: RNA sequencing, MEA: multielectrode array, PSCs: pluripotent stem cells.

Type of Device	References	Structural Visualization	Functional Activity Analysis	Transcriptomic Analysis	Proteomic Analysis	Metabolomic Analysis	Cellular Viability Analysis
Microfluidic devices with 3D culture areas and channels	[35,58]: Perfusable device with culture channels + perfusion channels Prenatal nicotine exposure	IF staining (on organoid cryosections)	Ø	RT-qPCR	Ø	Ø	TUNEL assay (on organoid cryosections)
	[59]: Device modeling cerebrospinal fluid flow + with brain ECM	IF staining (on organoid cryosections) + image-based quantifications	- Ca^2+^ imaging - Whole-cell patch-clamp	- RT-qPCR - RNAseq	LC-MS	- Glucose + lactate measurement in sampled culture medium - Computational simulation of glucose concentration	- Measurement of oxygen level within the organoids with oxygen-sensing nanoparticles (phosphorescence) - Necrotic area measurement (fluorescence)
	[60]: Device with constrained culture chamber Cerebral folding and wrinkles Lissencephaly modelling	- IF staining (directly in the device) - Fluorescent reporters used to electroporate PSCs (chromosomes + actin) (visualization directly in the device)	Ø	- RNAseq (RNA extracted directly from the device)	Ø	Ø	Ø
	[61]: Device to develop vascularized cerebral organoids	IF staining (directly in the device)	Ø	RT-qPCR	Ø	Ø	Ø
Microfluidic devices with micropillar arrays	[62,63]: Prenatal cadmium exposure	IF staining (on organoid cryosections)	Ø	RT-qPCR	Ø	Ø	Ø
	[64,65]: Prenatal exposure with valproic acid and breast cancer-derived exosomes	IF staining (on organoid cryosections) + image-based quantifications	Ø	- RT-qPCR - RNAseq	Ø	Ø	Cellular viability + hypoxia (directly in the device)
Microfluidic devices with air-liquid interface	[36]: Prenatal cannabis exposure	IF staining (on organoid cryosections)	2D MEA	RT-qPCR	Ø	Ø	Ø

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
