# Peer review of "Human Brain Organoids-on-Chip: Advances, Challenges, and Perspectives for Preclinical Applications"

_pharmaceutics, 2022, doi:10.3390/pharmaceutics14112301_

Round 1

Reviewer 1 Report

The manuscript by Castiglione et al. describes a review study on the developments in the field of human brain organoid on chip. The manuscript engagingly covers a wide range of literature with well-identified aspects including types of microfluidic settings, pros and cons of each type, commercialization perspectives, and more. I believe that this study would be of interest to the researchers working on brain on chip field as well as the ones looking for strategies for ADME experiments. Therefore, I suggest the acceptance of this study, yet have minor remarks to be added to the manuscript:

-       Development of the strategy also involves about data collection methods in these platforms. The review is much stronger if the authors amend it with comments regarding the applied data collection methods in these platforms.

-       MEA measurements are the workhorse of the BoC functions. This part can be extended a bit with more key examples.

-       Please add an explanation of the Lancaster method.

-       Please add representative images of the three microfluific settings.

-       Table 2 starts with references, but it is difficult to go back and forth. Please add representative/short titles of the references, so the reader decides to go to them or not immediately.

-       In this field, neuro-morphogenesis is also a major topic. It would be nice to involve it briefly in the manuscript if possible.

Author Response

-       Development of the strategy also involves about data collection methods in these platforms. The review is much stronger if the authors amend it with comments regarding the applied data collection methods in these platforms.

Additional information about read-outs and data collection methods identified in current Brain Organoids-on-Chips systems were added to the manuscript: and additional Table #3, as well as a full paragraph from line 647 to 713 that includes some perspectives.

-       MEA measurements are the workhorse of the BoC functions. This part can be extended a bit with more key examples.

We have added a paragraph to better describe this methodology applied to microfluidics from line 723 to 768

-       Please add an explanation of the Lancaster method.

To enhance the comprehension to a broader audience we have added the paragraph “2.2. Cerebral organoids: a diversity of generation methodologies terminology and classification “on line 170 to 178. This includes details on the terminology, as suggested by Pasca et al. in a very recent commentary published in Nature We hope this will suffice, as the Lancaster method has been extensively documented in the literature and referred to in our review.  

-       Please add representative images of the three microfluidic settings.

We have designed and added an illustration on to Figure 2 with a corresponding legend and updated manuscript accordingly.

-       Table 2 starts with references, but it is difficult to go back and forth. Please add representative/short titles of the references, so the reader decides to go to them or not immediately.

We have added explanations to Table 2 to better explicit referencing.

-       In this field, neuro-morphogenesis is also a major topic. It would be nice to involve it briefly in the manuscript if possible.

We have updated the manuscript as follows:

Line 80 “insights into cerebral organogenesis, functions and neurological disorders »,

Line 125 «2.1  Cerebral Brain organoids: promising in vitro 3D models of human brain organogenesis »

Paragraph line 139 to 155  

Reviewer 2 Report

Brain Organoids-on-Chips are considered as the promising platforms for pharmacological applications. The promising advantages of cerebral organoids for neurological disease models and their combination with microfluidic systems are summarized here, which will be helpful for the future research of neurological disorders. However, there are some questions to be elucidated.

1.     The construction methods of the neurological diseases should be illustrated in details, such as AD, PD, multiple sclerosis, etc.

2.     It is suggested that it should be supplemented with the commonly used cells to construct brain organoids-on-chips, such as neural mesenchymal stem cells, gliocytes.

3.     The representative biomarkers about the successful construction of brain organoids-on-chips should be supplemented.

It is suggested acceptance after revisions.

Author Response

Reviewer 2:

Brain Organoids-on-Chips are considered as the promising platforms for pharmacological applications. The promising advantages of cerebral organoids for neurological disease models and their combination with microfluidic systems are summarized here, which will be helpful for the future research of neurological disorders. However, there are some questions to be elucidated.

  1. The construction methods of the neurological diseases should be illustrated in detail, such as AD, PD, multiple sclerosis, etc.

We have added specific applications with corresponding references to illustrate the construction methods (lines 201 to 215). However, a full review of such application falls beyond the scope of this review whose purpose is to focus on microfluidic based organoids culture methods.

  1. It is suggested that it should be supplemented with the commonly used cells to construct brain organoids-on-chips, such as neural mesenchymal stem cells, gliocytes.

To take into account the possibility of adding cellular complexity to organoid culture, we have updated the manuscript as follows:

Line 80 “insights into cerebral organogenesis, functions and neurological disorders »,

Line 125 «2.1 Cerebral Brain organoids: promising in vitro 3D models of human brain organogenesis »

Paragraph line 139 to 155 and line 786 to 789.

  1. The representative biomarkers about the successful construction of brain organoids-on-chips should be supplemented.

We thank the reviewer for suggesting this – we have added relevant data in order to strengthen the message of the review. We have added a full paragraph from and from line 195 to 212 and line 707 to 711.
